# MULTIPLE MODES FOR CONTINUAL LEARNING

## ABSTRACT

Adapting model parameters to incoming streams of data is a crucial factor to deep learning scalability. Interestingly, prior continual learning strategies in online settings inadvertently anchor their updated parameters to a local parameter subspace to remember old tasks, else drift away from the subspace and forget. From this observation, we formulate a trade-off between constructing multiple parameter modes and allocating tasks per mode. Mode-Optimized Task Allocation (MOTA), our contributed adaptation strategy, trains multiple modes in parallel, then optimizes task allocation per mode. We empirically demonstrate improvements over baseline continual learning strategies and across varying distribution shifts, namely sub-population, domain, and task shift.

## 1 INTRODUCTION

As the world changes, so must our models of it. The premise of continual (or incremental or lifelong) learning is to build adaptive systems that enable a model to return accurate predictions as the test-time distribution changes, such as a change in domain or task. Training sequentially on multiple different task distributions tends to result in catastrophic forgetting (McCloskey & Cohen, 1989), where parameter updates benefiting the inference of the new task may worsen that of prior tasks. Alleviating this is the motivation for our work.

To enable flexibility in adoption, we do not assume parameter adaptation with respect to task-specific information, and we assume only access to model parameters alone (no conditioning inputs, query sets, rehearsal or replay buffers, $N$-shot metadata, or any historical data). This carries positive implications for adoption in online learning settings, and robustness towards different distribution shifts (e.g. sub-population, domain, task shifts). Interestingly, prior work in non-rehearsal methods (notably regularization and parameter isolation methods) tend to "anchor" the parameter updates with respect to a local parameter subspace. These methods begin with a model initialization, then update the model with respect to the first task, and henceforth all future parameter updates on new tasks are computed with respect to this local subspace (usually minimizing the number of parameter value changes). The key question we ask here: *what happens when we consider the global geometry of the parameter space?*

Our pursuit of an adaptation method leveraging global geometry is supported by various initial observations. When learning tasks $1, ..., T$, a multi-task learner tends to drift a large distance away from its previous parameters optimized for $1, ..., T-1$, indicating that when given information on all prior tasks, a multi-task learner would tend to move to a completely different subspace (Figure 3; Mirzadeh et al. (2020)). Catastrophic forgetting is intricately linked to parameter drift: unless drifting towards a multi-task-optimal subspace, if the new parameters drift further from the old parameter subspace, then accuracy is expected to drop for all prior tasks; not drifting sufficiently will retain performance on prior tasks, but fail on the new task. Coordinating parameter updates between multiple parameter modes tend to keep the average parameter drift distance low (Figure 3).

**Contributions.** Grounded on these findings, we introduce a new rehearsal-free continual learning algorithm (Algorithm 1). We initialize pre-trained parameters, maximize the distance between the parameters on the first task, then on subsequent tasks we optimize each parameter based on the loss with respect to their joint probability distribution as well as each parameter's drift from its prior position (and reinforce with backtracking). Evaluating forgetting per capacity, MOTA tends to outperform baseline algorithms (Table 3), and adapts parameters to sub-population, domain, and task shifts (Tables 1, 2).

**Related Work.** Lange et al. (2019) taxonomized continual learning algorithms into replay, regularization, and parameter isolation methods. Replay (or rehearsal) methods store previous task samples to supplement retraining with the new task, such as iCaRL (Rebuffi et al., 2017), ER (Ratcliff, 1990; Robins, 1995; Riemer et al., 2018; Chaudhry et al., 2019), and A-GEM (Chaudhry et al., 2018b). Regularization methods add regularization terms to the loss function to consolidate prior task knowledge, such as EWC (Kirkpatrick et al., 2017b), SI (Zenke et al., 2017), and LwF (Li & Hoiem, 2016). These methods tend to rely on no other task-specific information or supporting data other than the model weights alone. Parameter isolation methods allocate different models or subnetworks within a model to different tasks, such as PackNet (Mallya & Lazebnik, 2017), HAT (Serrà et al., 2018), SupSup (Wortsman et al., 2020), BatchEnsemble (Wen et al., 2020), and WSN (Kang et al., 2022). Task oracles may be required to activate the task-specific parameters. Ensembling strategies in this category may either require task indices to switch to a specific task model (e.g. Wen et al. (2020)), or update all ensemble models on all tasks but risk losing task-optimal properties of each parameter's subspace (e.g. Doan et al. (2022)).

The loss landscape changes after each task (Figure 3). Prior work either anchors to the local subspace of the first task, anchors each task to its specific local subspace, or anchors the entire parameter space to the last seen task. We are the first to leverage the global geometry of a loss landscape changing with tasks without compromising the task-optimal properties of each subspace nor requiring any task-specific information.

## 2 TRADE-OFF BETWEEN MULTIPLE MODES AND TASK ALLOCATION

First we introduce the problem set-up of continual learning, with assumptions extendable to broader online learning settings. Then we share the observations that motivate our study into multiple modes. Finally we present a trade-off, which motivates our proposed learning algorithm.

**Problem Setup.** A base learner receives $T$ tasks (or batches) sequentially. $\mathcal{D}_t = \{x_t, y_t\}$ denotes the dataset of the $t$-th task. In the continual learning setting, given loss function $\mathcal{L}$, a neural network $f(\theta; x)$ optimizes its parameters $\theta$ such that it can perform well on the $t$-th task while minimizing performance drop on the previous $(t-1)$ tasks: $\theta^* := \arg\min_\theta \sum_{t=1}^T \mathcal{L}(f(\theta; x_t), y_t)$. We assume the only information available at test-time is the model(s) parameters and the new task's data points. The learner cannot access any prior data points from previous tasks, and capacity is not permitted to increase after each task. Additionally, we do not assume parameter adaptation at test-time can be conditioned on task boundaries or conditioning inputs (task index, replay, $K$-shot query data, etc).

**Trade-off.** Many regularization-based methods are grounded on minimizing drift (change in parameters) to reduce forgetting on prior tasks. Yet in Table 3, a multi-task learner has a higher average drift consistently between tasks than EWC, even when both begin from a shared starting point (init $\rightarrow$ task 1). Given visibility to prior tasks, a multi-task learner will depart the subspace of the previous parameter, and drift far. This contradicts with the notion of forcing parameters to reside in the subspace of the previous parameter. The regularization-based methods essentially anchor all future parameters to the first task observed.

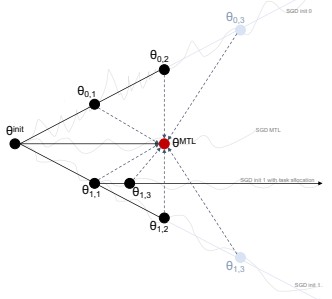

Results in mode connectivity (Garipov et al., 2018; Fort & Jastrzebski, 2019; Draxler et al., 2019) show that a single task can have multiple parameters ("modes") that manifest functional diversity. We explored computing multiple modes with respect to task 1 to incorporate the broader geometry of the parameter space beyond the subspace of one mode. To bring performance gains and capacity efficiency, we further obtained this trade-off between the number of modes, number of tasks allocated per mode, and capacity (Theorem 1).

Figure 1: A diagram of the different parameter trajectories demonstrating that, rather than anchoring all subsequent learning on mode $\theta_{0,1}$, we can leverage the functional diversity of other modes for optimal task allocation.

We denote $\theta^{\text{init}}$ as the initialization parameter, $\theta^{\text{MTL}(1,...,T)}$ as the multi-task parameter trained on tasks $1, ..., T$, and $\theta_{i,t}$ as the parameter of mode index $i$ updated on task $t$.

**Theorem 1** *If the number of modes $N$ is optimized against capacity $|\theta|$ and the set of tasks allocated per mode $|T(i) = \{t\}|$ for $i \in N$, $t \in T$, then the total task drift is lower in the multi-mode setting than single-mode setting:*

$$\Sigma_{i=1}^{N} \Sigma_{t}^{T(i)} \underbrace{\frac{1}{|\theta|/N} \Sigma_{d=1}^{|\theta|/N} (\theta_{i,t,d} - \theta_d^{\mathrm{MTL}})^2}_{\text{drift (multi-mode)}} < \Sigma_{t=2}^{T} \underbrace{\frac{1}{|\theta|} \Sigma_{d=1}^{|\theta|} (\theta_{1,t,d} - \theta_d^{\mathrm{MTL}})^2}_{\text{drift (single-mode)}}$$

*Proof.* See Appendix A.3.

As a result, our proposed algorithm is motivated to train multiple modes while optimizing for tasks learnt per mode. Our total capacity cannot exceed that of our baselines. Our improved performance in the continual learning setting also empirically validates Theorem 1.

## 3 MODE-OPTIMIZED TASK ALLOCATION

**Mode-Optimized Task Allocation (MOTA)** To implement our method, there are two components: (i) `initialize_parameters` initializing $N$ modes/parameters, and (ii) `update_parameters` updating a subset of modes/parameters. Once the first task is received, we train $N$ models in parallel such that the distance between them is maximized. Then for each subsequent task, we coordinate the parameter updates of the modes such that the drift per mode is minimized, and only the minimum number of modes needed to solve the task will be updated.

### 3.1 MODE INITIALIZATION

We begin with a pre-trained initialization (used for MOTA as well as all baselines). We instantiate $N$ models on this initialization of a fixed model architecture, and denote this set of parameters $\{\theta_{i,t}\}^N$ for $i \in N$ and $t \in T$. We train the parameters in parallel such that the distance between each other is maximized. For each batch per epoch, we randomly generate weights $\{\alpha_i\}^N$ that sum to 1 and compute an interpolated parameter $\widehat{\theta} = \Sigma_i^N \alpha_i \theta_{i,t}$. We compute the distance between the $N$ modes $\Sigma_{j,j \neq i}^N \mathtt{dist}(\theta_{i,t}, \theta_{j,t})$ to be maximized (adjusted with a penalty coefficient $\beta_{\max}$). We update each mode with respect to the input loss (evaluated with $\widehat{\theta}$) and the distance maximization term. For this distance maximization procedure, we used the average cosine similarity between each layer $\ell$ between every pair of models $\mathtt{dist} = \frac{1}{\frac{M}{2}(N^2 - N)} \Sigma_\ell^M \Sigma_{i=1}^{N-1} \Sigma_{j=i+1}^{N} \frac{\theta_i^\ell \cdot \theta_j^\ell}{||\theta_i^\ell|| \, ||\theta_j^\ell||}$, out of $M$ and $N$ layers and models respectively. Our coordination of distance maximization between a set of parameters is in-line with the methodology in Wortsman et al. (2021) and Datta & Shadbolt (2022), though their cases specify a unique random initialization per mode.

### 3.2 MODE ADAPTATION

The objective is to strategically update the parameters of a subset of modes required at the $t$-th task such that we minimize the overall drift from each mode's prior state but infer accurately on the task $t$. For each mode per epoch, we compute the loss with respect to a joint probability distribution. We also compute a distance term between each mode's parameters and its parameters at the $(t-1)$-th task $\mathtt{dist}(\theta_{i,t}, \theta_{i,t-1})$ to be minimized (adjusted with a penalty coefficient $\beta_{\min}$). We use the EWC regularization term for distance minimization. We update each mode with respect to the joint loss and its respective parameter drift, and checkpoint each update. We iterate through the modes sequentially per epoch to minimize memory requirements for parallelized training.

We compute the gradient update for each mode with respect to the joint probability distribution between all the modes. Specifically, we compute the average probability distribution returned at the last (softmax) layer $\ell = -1$ of each model $\rho_{\{\theta_{i,t}\}^N} = \frac{1}{N} \Sigma_i^N f(\theta_{i,t}^{\ell=-1}; x)$. If a task has a high level of certainty, then only a small subset of models would need to be updated and return a probability distribution skewed toward the target class while the other non-updated / minimally-updated models would return a random distribution, and the resulting averaged distribution would still be slightly skewed towards the target class. For a task of low certainty, then more models (of high functional diversity) would be updated to return a robust probability distribution. Furthermore, ensemble learning usually requires each ensemble model be trained independently (and usually with a different

---

**Algorithm 1** `update_parameters`

---

1: **procedure** update_parameters($\mathcal{D}_t, \{\theta_{i,t}\}_{i=1}^N$)  $\triangleright$ Pass a new task $\mathcal{D}_t$ to our current parameters $\{\theta_{i,t}\}_{i=1}^N$
2:     **if** $t=1$ **then**  $\triangleright$ Check if initializing parameters for the first time
3:         $\{\theta_{i,t}\}_{i=1}^N \leftarrow$ initialize_parameters($\mathcal{D}_1, \{\theta_{i,t}\}_{i=1}^N$)
4:     **else**
5:         $\{\theta_{i,t-1}\}_{i=1}^N \leftarrow \{\theta_{i,t}\}_{i=1}^N$  $\triangleright$ Retain a copy of the last task's parameters
6:         **for** $e$ in epochs **do**
7:             **for** $\theta_{i,t}$ in $\{\theta_{i,t}\}_{i=1}^N$ **do**
8:                 **for** $(x_t, y_t)$ in $\mathcal{D}_t$ **do**
9:                     $\rho_{\{\theta_{i,t}\}_{i=1}^N} =$ joint_inference($x_t, \{\theta_{i,t}\}_{i=1}^N$)
10:                     $\mathcal{L}_t = \mathcal{L}(\rho_{\{\theta_{i,t}\}^N}, y_t) + \beta_{\min} \operatorname{dist}(\theta_{i,t}, \theta_{i,t-1})$  $\triangleright$ Compute loss w.r.t. joint probability and drift
11:                     $\theta_{i,t} := \theta_{i,t} - \frac{\partial \mathcal{L}_t}{\partial \theta_{i,t}}$  $\triangleright$ Update each parameter independently
12:         $\{t^*, e^*\}^N := \arg\min_{\{t,e\}^N} \mathcal{L}(\rho_{\{\theta_{i,t,e}\}^N}, y_t)$  $\triangleright$ Backtracking: Enumerate through parameter combinations

$$+ \Sigma_{i=1}^N \operatorname{dist}(\theta_{i,t,e}, \theta_{i,t-1}) \text{ for } \{\theta_{i,t,e}\}_{i=1}^N \sim \{\theta_{i,t,e}\}_{i=1}^{N \times \text{epochs}}$$

13:         $\{\theta_{i,t}\}_{i=1}^N \leftarrow \{\theta_{i,t^*,e^*}\}_{i=1}^N$
14:     **return** $\{\theta_{i,t}\}_{i=1}^N$

---

**Algorithm 2** `initialize_parameters`

---

1: **procedure** initialize_parameters($\mathcal{D}_1, \{\theta_{i,t}\}_{i=1}^N$)  $\triangleright$ Initialize with task $\mathcal{D}_1$ and empty parameters set $\{\theta_{i,t}\}_{i=1}^N$
2:     $\{\theta_{i,t}\}_{i=1}^N \leftarrow \{\theta^{\text{init}}\}_{i=1}^N$
3:     **for** $e$ in epochs **do**
4:         **for** $(x_t, y_t)$ in $\mathcal{D}_1$ **do**
5:             $\alpha_i \sim [0,1]\ \forall i \in N$ s.t. $\Sigma_{i=1}^N \alpha_i \equiv 1$
6:             $\widehat{\theta} = \Sigma_{i=1}^N \alpha_i \theta_{i,t}$  $\triangleright$ Sample interpolated parameter $\widehat{\theta}$
7:             $\mathcal{L}_t = \mathcal{L}(f(\widehat{\theta}); x_t, y_t) + \beta_{\max} \Sigma_{j=1, j\neq i}^N \operatorname{dist}(\theta_{i,t}, \theta_{j,t})$  $\triangleright$ Compute loss and distance term
8:             $\theta_{i,t} := \theta_{i,t} - \frac{\partial \mathcal{L}_t}{\partial \theta_{i,t}}$  $\triangleright$ Update each parameter independently
9:     **return** $\{\theta_{i,t}\}_{i=1}^N$

---

**Algorithm 3** `joint_inference`

---

1: **procedure** joint_inference($x, \{\theta_{i,t}\}_{i=1}^N$)  $\triangleright$ Inference using the set of parameters $\{\theta_{i,t}\}_{i=1}^N$
2:     $\rho_{\{\theta_{i,t,e}\}^N} = \frac{1}{N}\Sigma_{i=1}^N f(\theta_{i,t}^{\ell=-1}; x)$  $\triangleright$ Taking average of the joint probability distribution returned at the softmax layer $\ell = -1$
3:     **return** $\rho_{\{\theta_{i,t,e}\}^N}$

---

initialization) and the final predictions are obtained as an average of the predictions of each model. Averaging procedures during inference (e.g. averaging activations, averaging softmax) are not used during training, as each ensemble model should not influence the gradient computation of another, and all ensemble models are expected to be trained on all tasks.

We checkpoint gradient updates per epoch, and can thus further optimize the sequence of mode updates by backtracking. Despite optimizing with respect to the joint probability distribution, one risk remains where a minimum number of gradient updates across all modes need to take place before a stable joint probability distribution can be computed. In other words, by the time we have jointly-accurate modes, it is likely most of the modes have been over-optimized, and thus drifted more than needed. Thus, we need to backtrack and find the optimal checkpoints across modes that minimize loss with respect to their checkpointed joint probability distribution and drift. We adopted the simplest backtracking algorithm: we enumerate through every combination of model checkpoint per epoch across the modes, and select the checkpoint combination that minimizes the loss with respect to the joint probability distribution for task $t$ and minimizes total parameter drift. To reduce the propensity of selecting earlier-epoch checkpoints (e.g. non-updated models), we can add a penalty term to distance regularization to reduce its relative weighting to input loss. This helps mitigate the loss imbalance between the input loss term and distance term for earlier checkpoints.

## 4 EVALUATION

We state our experimental setup below, with configuration details in Appendix A.1. We then review our results on MOTA's improvement in task adaptation.

**Incremental Learning (IL) Settings.** A *task* is a subsequent training phase with a new batch of data, pertaining to a new sub-population/domain, new label set, or different output space. In instance-IL, each new task bring new instances from known classes. In class/task-IL, each new task bring instances from new classes only. Class-IL performs inference w.r.t. all observed classes. Task-IL performs inference w.r.t. the label set of the task. We evaluate on task-IL unless otherwise specified.

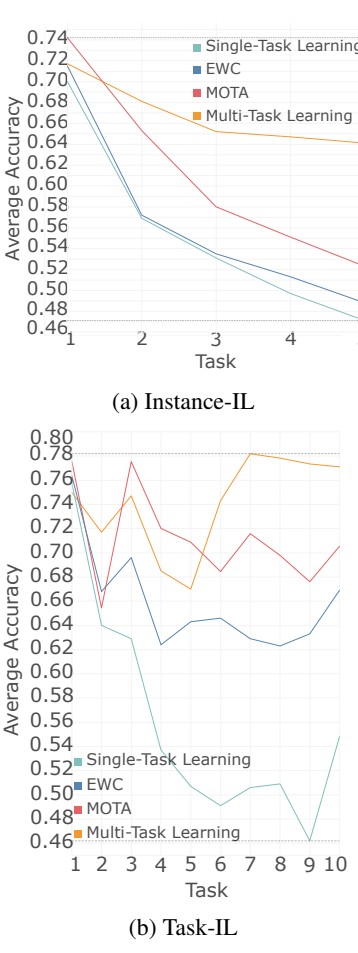

(a) Instance-IL

(b) Task-IL

Figure 1: Per-task average accuracy

**Datasets.** Task-IL Split-CIFAR100 (Krizhevsky, 2009) is constructed by dividing 100 fine labels into 10 tasks (10 fine labels per task). Instance-IL Split-CIFAR100 (Krizhevsky, 2009) is constructed by dividing 100 fine labels (mapped to 5 coarse labels) into 5 tasks (20 coarse labels per task). Instance-IL DomainNet (Peng et al., 2019) is composed of 6 domains with 345 labels. Task-IL TinyImageNet (Stanford) is constructed by dividing 200 labels into 10 tasks (20 labels per task).

**Architectures.** We initialize ResNets (He et al., 2015), loading ImageNet-pretrained weights from PyTorch. We train ResNet-18 ($11, 181, 642$ parameters), ResNet-50 ($23, 528, 522$ parameters), and ResNet-152 ($58, 164, 298$ parameters). To retain comparable capacity, ResNet-50 (-18) is the default model for baselines (MOTA). We do not assume task boundaries at test-time for parameter adaptation, and do not use task index to recompute task-specific parameters. We evaluate against regularization methods (Elastic Weight Consolidation (Kirkpatrick et al., 2017b), Synaptic Intelligence (Zenke et al., 2017), Learning without Forgetting (Li & Hoiem, 2016)). We also compare against four replay baselines (Experience Replay (Rolnick et al., 2018), Averaged Gradient of Episodic Memory (Chaudhry et al., 2018b), Gradient Projection Memory (Saha et al., 2021), and La-MAML (Gupta et al., 2020)), though they require a task replay buffer for parameter adaptation. We further evaluate against two parameter isolation baselines (BatchEnsemble (Wen et al., 2020), and Winning SubNetworks (Kang et al., 2022)). We introduce two ablations: Ensemble (distance max.) which is an ensemble of modes obtained using MOTA's distance maximization procedure and trained on all tasks, and Ensemble (independent seeds) which is an ensemble of modes trained on all tasks but from independent random initializations.

**Metrics.** Single-Task Learning trains on each task independently. Multi-Task Learning trains on all seen tasks simultaneously. Primarily baselined against regularization methods, capacity is the number of trainable model parameters. When considering replay methods as well, we distinguish capacity w.r.t. model parameters from repay buffer, where the replay budget is 100 per class. The average drift distance between tasks is the distance between the updated parameters and previous parameters, averaged for each task update instance. For multiple model parameters, we take the cumulative distance. We compute this from $\frac{1}{T-1}\Sigma_{t=2}^{T}\Sigma_i^N \text{dist}(\theta_{i,t}, \theta_{i,t-1})$. Given $\text{Acc}(\theta, x_t)$ as the validation accuracy on the $t$-th task, average accuracy is the average validation accuracy across all seen tasks w.r.t. the parameters updated at the $t$-th task. We compute this from $\frac{1}{t} \sum_{v=1}^{t} \text{Acc}(\theta_t, x_v)$. Backward Transfer (Lopez-Paz & Ranzato, 2017) measures the influence that learning a task has on the performance on previous tasks. We compute this from $\frac{1}{t-1} \sum_{v=1}^{t-1} \text{Acc}(\theta_t, x_v) - \text{Acc}(\theta_v, x_v)$. Forward Transfer (Lopez-Paz & Ranzato, 2017) measures the influence that learning a task has on the performance of future tasks. We compute this from $\frac{1}{t-1} \sum_{v=2}^{t} \text{Acc}(\theta_{v-1}, x_v) - \text{Acc}(\theta^{\text{init}}, x_v)$. Remembering (Díaz-Rodríguez et al., 2018) computes the forgetting part of Backward Transfer. We compute this

from $1 - |\min(0, \frac{1}{t-1} \sum_{v=1}^{t-1} \mathrm{Acc}(\theta_t, x_v) - \mathrm{Acc}(\theta_v, x_v))|$. Forgetting (Chaudhry et al., 2018a) is calculated by the difference of the peak accuracy and ending accuracy of each task. We compute this from $\frac{1}{T-1} \sum_{v=1}^{T-1} \max_{t \in \{1,\dots,T-1\}} (\mathrm{Acc}(\theta_t, x_v) - \mathrm{Acc}(\theta_T, x_v))$.

## 4.1 EVALUATING CATASTROPHIC FORGETTING

We evaluate MOTA against different types of distribution shifts (Table 1, Figure 1, Table 2). Evaluating on task shift in CIFAR100 and TinyImageNet, we observe improved backward and forward transfer with MOTA, indicating lower forgetting as well as improved feature transferability between tasks. As task-specific information is not necessary for parameter adaptation, we can evaluate our method on settings that do not require an assumption of task boundaries or indexing, namely sub-population and domain shift. We find that our method also outperforms on backward/forward transfer on Instance-IL CIFAR100 (sub-population shift) and Task-IL DomainNet (domain shift).

We primarily baseline MOTA against other regularization-based methods (Table 3). Though MOTA aims to find a combination of parameters that can perform close to the multi-task learning strategy, it falls shortly behind. By optimizing (i) the number of modes throughout the parameter space against (ii) optimal task allocation per parameter, MOTA can outperform other regularization and replay methods. In particular, MOTA outperforms its component baselines: EWC (where this baseline and MOTA use the EWC regularization term in minimizing mode drift), and ensembling (where this baseline and MOTA use the same distance maximization procedure from init to return modes). We show that the optimal combination of these components can yield superior performance.

Further evaluating MOTA against ensembles, an ablation multi-mode strategy where each ensemble mode is sequentially-trained on all tasks without any forgetting strategies, we find that ensembles underperform most baselines and MOTA. Ensembling with independent seeds is almost equivalent in performance to single-task training. Though ensembling maximizes functional diversity across

Table 1: **Metrics evaluation**: We evaluate distance, capacity, and other forgetting measures on Split-CIFAR100. Instance-IL (a) presumes the coarse labels are the same between task (5 tasks, 20 labels), thus is representative of sub-population shift. Task-IL (b) presumes unique fine labels per task (10 tasks, 10 labels), and is representative of the general continual learning setting. For all comparisons, average task drift begins from parameters updated on Task 1, not from initialization.

(a) Instance-IL

| Method | Average Accuracy ↑ | Average Task Drift ↓ | Capacity ↓ | Backward Transfer ↑ | Forward Transfer ↑ | Remembering ↑ | Forgetting ↓ |
|---|---|---|---|---|---|---|---|
| Single-Task Learning | 47.1 | 1.0 | 23,528,522 | -20.2 | 40.8 | 79.8 | 27.0 |
| EWC | 48.8 | 0.360 | 23,528,522 | -12.7 | 47.2 | 87.3 | 17.0 |
| MOTA | 52.3 | $3.13 \times 10^{-5}$ | 22,363,284 | -11.6 | 55.4 | 88.4 | 13.9 |
| Multi-Task Learning | 64.1 | 0.910 | 23,528,522 | 0 | 0 | 1 | 0 |

(b) Task-IL

| Method | Average Accuracy ↑ | Average Task Drift ↓ | Capacity ↓ | Backward Transfer ↑ | Forward Transfer ↑ | Remembering ↑ | Forgetting ↓ |
|---|---|---|---|---|---|---|---|
| Single-Task Learning | 54.8 | 1.0 | 23,528,522 | -24.7 | 46.9 | 75.3 | 23.7 |
| EWC | 66.9 | 0.521 | 23,528,522 | -8.03 | 62.7 | 92.0 | 6.69 |
| MOTA | 70.5 | $7.85 \times 10^{-6}$ | 22,363,284 | -5.08 | 69.5 | 94.9 | 2.89 |
| Multi-Task Learning | 77.1 | 0.725 | 23,528,522 | 0 | 0 | 1 | 0 |

Table 2: **Varying Datasets:** We continue our Task-IL evaluation on another task shift dataset (a), and domain shift dataset (b).

(a) TinyImageNet

| Method | Average Accuracy ↑ | Backward Transfer ↑ | Forward Transfer ↑ | Remembering ↑ | Forgetting ↓ |
|---|---|---|---|---|---|
| Single-Task Learning | 65.2 | -14.7 | 65.0 | 85.3 | 16.5 |
| EWC | 76.7 | -4.49 | 74.7 | 95.5 | 3.45 |
| MOTA | 82.7 | -1.70 | 81.2 | 98.3 | 1.57 |
| Multi-Task Learning | 90.2 | 0 | 0 | 1 | 0 |

(b) DomainNet

| Method | Average Accuracy ↑ | Backward Transfer ↑ | Forward Transfer ↑ | Remembering ↑ | Forgetting ↓ |
|---|---|---|---|---|---|
| Single-Task Learning | 34.8 | -17.5 | 22.2 | 82.5 | 27.1 |
| EWC | 40.2 | -9.82 | 31.6 | 90.2 | 8.57 |
| MOTA | 57.5 | -1.62 | 55.9 | 98.4 | 2.43 |
| Multi-Task Learning | 70.4 | 0 | 0 | 1 | 0 |

Table 3: **Baseline comparison:** Evaluating on Task-IL Split-CIFAR100, we evaluate MOTA against Single/Multi-Task Learning (the lower/upper bound), regularization (in-line assumptions) and replay / parameter isolation (more difficult to beat) methods, and ensemble ablations. We keep capacity at most $1\times$ ResNet50 for fair comparison. Baseline configurations are listed in Appendix A.1.

| Method | Average Accuracy ↑ | Storage: Model parameters ↓ | Storage: Replay buffer ↓ |
|---|---|---|---|
| Single-Task Learning | 54.8 | 23,528,522 | – |
| EWC (Kirkpatrick et al., 2017b) | 66.9 | $2 \times 23,528,522$ | – |
| SI (Zenke et al., 2017) | 63.7 | $3 \times 23,528,522$ | – |
| LwF (Li & Hoiem, 2016) | 61.2 | 23,528,522 | – |
| ER (Riemer et al., 2018) | 68.2 | 23,528,522 | 10,000 |
| A-GEM (Chaudhry et al., 2018b) | 67.2 | $2 \times 23,528,522$ | 10,000 |
| La-MAML (Gupta et al., 2020) | 65.8 | 23,528,522 | 10,000 |
| GPM (Saha et al., 2021) | 67.4 | 23,528,522 | 10,000 |
| BatchEnsemble (Wen et al., 2020) | 62.2 | 30,116,508 | – |
| WSN (Kang et al., 2022) | 68.7 | 23,199,123 | – |
| MOTA | 70.5 | $2 \times 11,181,642$ | – |
| Ensembles (distance max.) | 60.1 | $2 \times 11,181,642$ | – |
| Ensembles (independent seeds) | 55.5 | $2 \times 11,181,642$ | – |
| Multi-Task Learning | 77.1 | 23,528,522 | – |

modes, contains diverse representations per task, and increases the likelihood of a mode being closer to a multi-task parameter, it does not have a coordinated inference strategy (e.g. weighted mode predictions) nor is it capacity-efficient with respect to tasks (updating all modes or a subset of modes). Furthermore, ensembles are trained on all tasks and each mode returns an informed probability distribution conditioned on all tasks while MOTA dilutes the joint probability distribution with partially-conditioned probability distributions of updated models with random distributions of non-updated models, thus MOTA would be expected to underperform ensembling.

Unlike prior continual learning methods, we do not assume each parameter must have seen all prior tasks. We introduce efficiency in task allocation per parameter. This is similarly motivated to a multi-headed architecture (where distinct subnetworks are explicitly allocated to a different subset of tasks) with a shared header, though we do not need a task index to select a task-specific subnetwork.

## 4.2 TRADEOFF BETWEEN ACCURACY AND CAPACITY

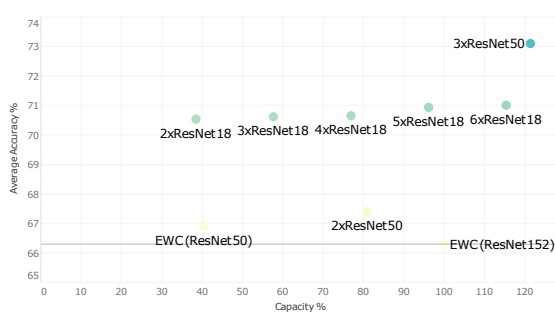

Figure 2: **Varying modes count:** Compared to EWC (ResNet50/152), we evaluate the trade-off between accuracy against capacity (number of modes), given a constant number of tasks.

In Figure 2, we vary the capacity as the number of total trainable parameters in proportion to the number of parameters for EWC (using a ResNet152 model). $2\times$ ResNet18 is similar in capacity to EWC (ResNet50), and the other architectures of varying number of modes have less than or similar capacity to EWC (ResNet152).

We learn from ensemble performance in Table 3 that utilizing global geometry alone is not sufficient to improve average accuracy, and that we need to optimally allocate tasks per mode. Similarly in Figure 2, we observe a similar inclination for a balance between capturing global geometry and optimizing tasks per mode. We would expect that, given a constant number of tasks, an increasing number of modes would result in improved average accuracy. Instead, the average accuracy gain between 2-6 ResNet18s is minimal.

We also observe a trade-off between the number of modes and optimal task allocation per mode. Considering EWC (ResNet50; i.e. $1\times$ ResNet50) and $2-3\times$ ResNet50, an increase in the number of modes results in an increase in average accuracy. Considering constant capacity, $4\times$ ResNet18 outperforms $2\times$ ResNet50; however, $3\times$ ResNet50 outperforms $6\times$ ResNet18.

### 4.3 CHANGES TO THE GEOMETRY OF THE PARAMETER SPACE

From Table 1, the average task drift (drift distance between the next and previous task's parameters) tends to be lower for MOTA than EWC, Single-Task, and Multi-Task Learning. This can be visually observed in the trajectory of the parameters in Figure 3.

We also observe from Figure 3 that the loss landscape changes drastically between tasks. A region considered to be low-loss by a parameter at task $t$ becomes a high-loss region with respect to the next task $t + 1$. As each task is added, the sharpness of the basin upon which the EWC parameter exists tends to increase. This change in sharpness tends to be much smaller for the regions in which MOTA modes are located, where the basin still retains a similar level of flatness.

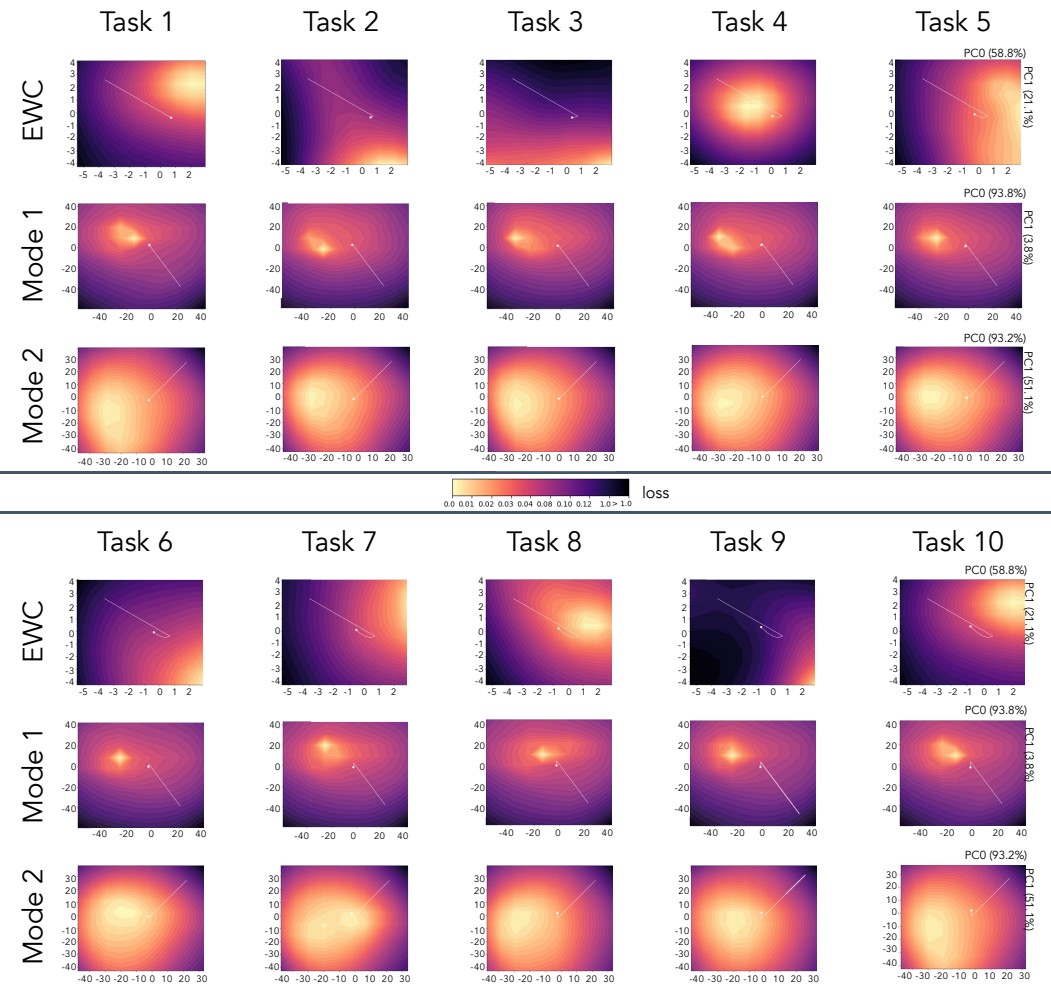

Figure 3: **Loss Landscape:** In-line with Li et al. (2018), we visualize the loss landscape by storing the model parameters along the optimization trajectory per epoch from Task 1-10 (including the last parameter $\theta^*$), identify the top two components/directions $\delta$, $\eta$ with PCA, and with respect to each task's dataset $x_t$, $y_t$ we plot the loss function $\mathcal{L}(\theta^* + \alpha_\delta \delta + \alpha_\eta \eta)$ with varying interpolation coefficients $\alpha_\delta$, $\alpha_\eta$. We plot each set per method across the tasks to show the relative change in flatness/sharpness between tasks. We normalize the loss values of all plots jointly between 0 and 1. The trajectory (white line) is the position of the parameter in the parameter space at the $t$-th task. Note that the loss values are not necessarily synchronized for each parameter between tasks (e.g. the init parameter) as the loss for the same parameter may be different for different tasks.

## 5 CONCLUSION

Driven by observations in the optimization behavior of multi-task learners, we hypothesize incorporating the broader geometry of the parameter space into a continual learner for improved adaptation between data distributions. Supported by the formulation of a trade-off between the number of modes and task allocation per mode, we demonstrate that Mode-Optimized Task Allocation (MOTA) can outperform existing baselines. It can retain a high average accuracy on current and previous data in sub-population, domain, and task shift settings. We also present supporting results on how MOTA influences the sharpness of the loss landscape between tasks, and how accuracy varies with the total capacity of MOTA. With this first step in leveraging the global geometry of loss landscapes changing with tasks, many potential future directions exist. Next steps range from more efficient methods in leveraging the global geometry (e.g. using fewer modes or a single network alone, alternative task allocation schemes), to demonstrating improved properties with architecture modifications (e.g. improve robustness with changes to regularization terms), to supporting meta learning regimes (e.g. unseen tasks).

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

# A  APPENDIX

This appendix is organized as follows:

- **Appendix A.1:** We provide detail on the experimental configurations.
- **Appendix A.2:** We review the EWC regularization term, given its regular usage as a baseline as well as its usage as a distance minimization term in MOTA.
- **Appendix A.3:** We provide our complete analysis for the trade-off between the number of modes and task allocation.

## A.1  EXPERIMENTAL CONFIGURATIONS

- **Training:** We loaded pre-trained ImageNet weights for each ResNet architecture with PyTorch (Paszke et al., 2019). We trained for 200 epochs, with batch size 512, using AdamW optimizer (learning rate 0.1 with 1cycle learning rate policy) train/val/test split of 70/10/20%, We train and evaluate (including when averaging the joint probability distribution) using a cross-entropy loss function. We used the seed 3407 throughout all operations; for those requiring multiple unique random values (e.g. multiple randomly initialized models), the seed is the index of the object (1, 2, ...).
- **Regularization methods:** The regularization strength for weight penalty $\lambda$ for EWC and SI is $1,000$ and $100$ respectively, with SI dampening term $0.1$, and LwF's temperature for distillation loss $2.0$.
- **Replay methods:** The memory buffer's budget per class is $100$. We use $10^{-7}$ A-GEM epsilon (parameter to ensure numerical stability of A-GEM). For La-MAML (Gupta et al., 2020), a meta-learning approach with a replay buffer, we retain the per-parameter learning rate $\alpha_0 = 0.1$ and the learning rate of $\alpha$ at $0.3$. We use $0.01$ learning rate for GPM (Saha et al., 2021).
- **Parameter isolation methods:** For WSN (Kang et al., 2022), which selectively reuses and expands subnetworks within a network, we set layer-wise capacity to $0.5$. BatchEnsemble (Wen et al., 2020) uses a base network (slow weights) and stores separate parameters (fast weights) to compute the parameters per ensemble, thus $N$ ensembles do not require $N$ sets of parameters. Each ensemble member is responsible for one task ($N = 10$). We retain -0.5 random sign init for fast weights and $0.5$ fast weights learning rate multiplier.
- **MOTA:** The distance maximization coefficient $\beta_{\max}$ is $100.0$. For the distance minimization procedure in subsequent epochs, we retain the elastic weights consolidation procedure of computing the Fisher information matrix and computing its corresponding regularization term. We retain EWC's lambda $\beta_{\min} = \lambda = 1,000$.
- **Ensembles methods:** We used the same number of models in ensemble as the number of modes of the comparable MOTA ($N = 2$ for Table 3). We retain the same distance maximization coefficients as MOTA, and use unique seeds (1,2,...) for each model's random initialization.

## A.2  EWC REGULARIZATION TERM

Importance of each parameter is computed for each task by the parameter's corresponding diagonal element from its Fisher Information matrix $F$. Given the index of the parameters $i$ ($i^{\text{th}}$ element of $\theta_t$, $i^{\text{th}}$ diagonal element of $F$), importance of the previous task compared to the next task $\lambda$, we can compute the EWC regularization term (Kirkpatrick et al., 2017a): $\sum_{i=1}^{|\theta_{t-1}|} \frac{\lambda}{2} F_i (\theta_t - \theta_{t-1})^2$.

## A.3  ANALYSIS ON MULTIPLE MODES VS TASK ALLOCATION TRADE-OFF

First we denote $\theta^{\text{init}}$ as the initialization parameter, $\theta^{\text{MTL}(1,...,\text{T})}$ as the multi-task parameter trained on tasks $1, ..., T$, and $\theta_{i,t}$ as the parameter of mode index $i$ updated on task $t$.

**Lemma 1** *Iterating through each task $t$, for a reference multi-task parameter $\theta^{\text{MTL}}$, the cumulative distance between an updated $\theta_{i,t}$ and previous $\theta_{i,t-1}$ parameter with respect to $\theta^{\text{MTL}}$ will exceed the*

*drift between $\theta_{i,t}$ and $\theta_{i,t-1}$:*

$$\Sigma_{t=2}^{T}(\theta_{i,t} - \theta^{\mathrm{MTL}}) > \Sigma_{t=2}^{T}(\theta_{i,t} - \theta_{i,t-1})$$

*Proof.* By the triangle inequality, the sum of the distances between the previous and the updated parameter with respect to $\theta^{\mathrm{MTL}}$ will exceed the drift between the previous and the updated parameter:

$$(\theta_{i,t} - \theta^{\mathrm{MTL}}) + (\theta_{i,t-1} - \theta^{\mathrm{MTL}}) > (\theta_{i,t} - \theta_{i,t-1}) \tag{1}$$

Thus, we can show that that the cumulative distance between an updated parameter with respect to $\theta^{\mathrm{MTL}}$ will exceed the drift between the updated and previous parameters. Hence, this cumulative distance also measures the task drift.

$$\Sigma_{t=2}^{T}(\theta_{i,t} - \theta^{\mathrm{MTL}}) > \Sigma_{t=2}^{T}(\theta_{i,t} - \theta_{i,t-1}) \tag{2}$$

$\square$

**Definition 1** (Task Allocation) *Task Allocation is defined as a procedure that allocates a set of tasks $T(i) = \{t\}$ to be learnt by a parameter mode $\theta_{i,t}$ of index $i$.*

> **Definition 1.1** (Optimal Task Allocation) *Optimal Task Allocation is defined as a procedure that maximizes the number of tasks $|\{t\}|$ allocated per mode of index $i$, while minimizing the total drift between parameter updates $\Sigma_{i=1}^{N}\Sigma_{t}^{T(i)}(\theta_{i,t} - \theta_{i,t-1})$.*
>
> **Corollary 1** *We can approximate Optimal Task Allocation by optimizing the number of tasks $|\{t\}|$ allocated to mode $i$ against the cumulative distance between an updated $\theta_{i,t}$ and previous $\theta_{i,t-1}$ parameter with respect to $\theta^{\mathrm{MTL}}$ (Lemma 1). This results in:*
>
> $$T(i) := \min \underset{|\{t\}| \geq 1}{\arg\max} \sum_{i=1}^{N} \sum_{t}^{T(i)} (\theta_{i,t} - \theta^{\mathrm{MTL}}) \tag{3}$$
>
> *In other words, $T(i) \propto \frac{1}{\theta_{i,t} - \theta^{\mathrm{MTL}}}$. At least one task must be allocated per mode $|T(i)| \geq 1$.*
>
> **Corollary 2** *Given $\mathcal{L}(f(\theta^{\mathrm{MTL}(1,...,T)}; x^{T(i)}), y^{T(i)}) \approx \mathcal{L}(f(\theta^{T(i)}; x^{T(i)}), y^{T(i)})$, we use $\theta^{\mathrm{MTL}} = \theta^{\mathrm{MTL}(1,...,T)}$ as the reference multi-task parameter.*

**Theorem 1** *If the number of modes $N$ is optimized against capacity $|\theta|$ and the set of tasks allocated per mode $|T(i) = \{t\}|$ for $i \in N$, $t \in T$, then the total task drift is lower in the multi-mode setting than single-mode setting:*

$$\Sigma_{i=1}^{N}\Sigma_{t}^{T(i)}\frac{1}{|\theta|/N}\Sigma_{d=1}^{|\theta|/N}(\theta_{i,t,d} - \theta_{d}^{\mathrm{MTL}})^2 < \Sigma_{t=2}^{T}\frac{1}{|\theta|}\Sigma_{d=1}^{|\theta|}(\theta_{1,t,d} - \theta_{d}^{\mathrm{MTL}})^2$$

*Proof.* Based on Lemma 1, given $N$ modes and optimal task allocation $T(i)$ with respect to the distance between each $\theta_i$ and $\theta^{\mathrm{MTL}}$, we can compute the total drift with respect to $\theta^{\mathrm{MTL}}$ as $\Sigma_{i=1}^{N}\Sigma_{t}^{T(i)}(\theta_{i,t} - \theta^{\mathrm{MTL}})$.

Note that the capacity of an evaluated mode changes between the multi-mode and single-mode setting. We can compute the total drift normalized by capacity (specifically the number of parameter values) with the squared Euclidean distance averaged by number of dimensions $\frac{1}{|\theta^{\mathrm{MTL}}|}\Sigma_{d=1}^{|\theta^{\mathrm{MTL}}|}(\theta_{i,t,d} - \theta_{d}^{\mathrm{MTL}})^2$, given $|\theta^{\mathrm{MTL}}| \equiv |\theta_{i,t}|$.

From Corollary 1, $|T(i)|$ is larger when $\theta_i$ is closer to $\theta^{\mathrm{MTL}}$. Thus for a threshold $\mathcal{T}$, we can decompose the total drift into:

$$\begin{aligned} &\Sigma_{i=1}^{N}\Sigma_{t}^{T(i)}\frac{1}{|\theta|/N}\Sigma_{d=1}^{|\theta|/N}(\theta_{i,t,d} - \theta_{d}^{\mathrm{MTL}})^2 \\ =&\Sigma_{i=1}^{N}\left[\Sigma_{t}^{T(i)\big||T(i)|>\mathcal{T}}\frac{1}{|\theta|/N}\Sigma_{d=1}^{|\theta|/N}(\theta_{i,t,d} - \theta_{d}^{\mathrm{MTL}})^2 + \Sigma_{t}^{T(i)\big||T(i)|\leq\mathcal{T}}\frac{1}{|\theta|/N}\Sigma_{d=1}^{|\theta|/N}(\theta_{i,t,d} - \theta_{d}^{\mathrm{MTL}})^2\right] \end{aligned} \tag{4}$$

Consequently, taking the difference in total drift (Eqt 4) between multiple-mode against single-mode settings result in the follow trade-off function (Eqt 5). We add $1/N$ to enable comparison between

single-mode and multi-mode, given their $N$ differs.

$$\min \pi = \Sigma_{i=1}^N \left[ \Sigma_t^{T(i)} \frac{1}{|\theta|/N} \Sigma_{d=1}^{|\theta|/N} (\theta_{i,t,d} - \theta_d^{\text{MTL}})^2 - \frac{1}{N} \Sigma_{t=2}^T \frac{1}{|\theta|} \Sigma_{d=1}^{|\theta|} (\theta_{1,t,d} - \theta_d^{\text{MTL}})^2 \right]$$

$$= \Sigma_{i=1}^N \left[ \Sigma_t^{T(i)\big||T(i)|>\mathcal{T}} \frac{1}{|\theta|/N} \Sigma_{d=1}^{|\theta|/N} (\theta_{i,t,d} - \theta_d^{\text{MTL}})^2 + \Sigma_t^{T(i)\big||T(i)|\leq\mathcal{T}} \frac{1}{|\theta|/N} \Sigma_{d=1}^{|\theta|/N} (\theta_{i,t,d} - \theta_d^{\text{MTL}})^2 \right]$$

$$- \frac{1}{N} \Sigma_{i=1}^N \left[ \Sigma_t^{T(i)\big||T(i)|>\mathcal{T}} \frac{1}{|\theta|} \Sigma_{d=1}^{|\theta|} (\theta_{1,t,d} - \theta_d^{\text{MTL}})^2 + \Sigma_t^{T(i)\big||T(i)|\leq\mathcal{T}} \frac{1}{|\theta|} \Sigma_{d=1}^{|\theta|} (\theta_{1,t,d} - \theta_d^{\text{MTL}})^2 \right] \tag{5}$$

Notably, for small $\mathcal{T}$ (e.g. $\mathcal{T} = 1$), the $\Sigma_t^{T(i)\big||T(i)|\leq\mathcal{T}} \frac{1}{|\theta|/N} \Sigma_{d=1}^{|\theta|/N} (\theta_{i,t,d} - \theta_d^{\text{MTL}})^2$ term only learns a few tasks per mode, lowers the capacity available per mode $|\theta|/N$, and thus these capacity-inefficient modes are *redundant*. Furthermore, as $\mathcal{T}$ decreases, the functional diversity of a mode is less important, and any random mode can generalize the set of tasks $T(i)\big||T(i)| \leq \mathcal{T}$. Hence,

$$\Sigma_t^{T(i)\big||T(i)|\leq\mathcal{T}} \frac{1}{|\theta|/N} \Sigma_{d=1}^{|\theta|/N} (\theta_{i,t,d} - \theta_d^{\text{MTL}})^2 \approx \Sigma_t^{T(i)\big||T(i)|\leq\mathcal{T}} \frac{1}{|\theta|} \Sigma_{d=1}^{|\theta|} (\theta_{1,t,d} - \theta_d^{\text{MTL}})^2 \tag{6}$$

If $N = 1$, then $\pi = 0$, given:

$$\Sigma_t^{T(i)\big||T(i)|>\mathcal{T}} \frac{1}{|\theta|/N} \Sigma_{d=1}^{|\theta|/N} (\theta_{i,t,d} - \theta_d^{\text{MTL}})^2 + \Sigma_t^{T(i)\big||T(i)|\leq\mathcal{T}} \frac{1}{|\theta|/N} \Sigma_{d=1}^{|\theta|/N} (\theta_{i,t,d} - \theta_d^{\text{MTL}})^2$$

$$\equiv \Sigma_t^{T(i)\big||T(i)|>\mathcal{T}} \frac{1}{|\theta|} \Sigma_{d=1}^{|\theta|} (\theta_{1,t,d} - \theta_d^{\text{MTL}})^2 + \Sigma_t^{T(i)\big||T(i)|\leq\mathcal{T}} \frac{1}{|\theta|} \Sigma_{d=1}^{|\theta|} (\theta_{1,t,d} - \theta_d^{\text{MTL}})^2 \tag{7}$$

Performance would be identical to the single-mode sequential learning case.

If $N \to \infty$ (and redundant modes dominate), then $\pi > 0$, given:

$$\Sigma_t^{T(i)\big||T(i)|>\mathcal{T}} \frac{1}{|\theta|/N} \Sigma_{d=1}^{|\theta|/N} (\theta_{i,t,d} - \theta_d^{\text{MTL}})^2 + \Sigma_t^{T(i)\big||T(i)|\leq\mathcal{T}} \frac{1}{|\theta|/N} \Sigma_{d=1}^{|\theta|/N} (\theta_{i,t,d} - \theta_d^{\text{MTL}})^2$$

$$> \frac{1}{N} \Sigma_t^{T(i)\big||T(i)|>\mathcal{T}} \frac{1}{|\theta|} \Sigma_{d=1}^{|\theta|} (\theta_{1,t,d} - \theta_d^{\text{MTL}})^2 + \Sigma_t^{T(i)\big||T(i)|\leq\mathcal{T}} \frac{1}{|\theta|} \Sigma_{d=1}^{|\theta|} (\theta_{1,t,d} - \theta_d^{\text{MTL}})^2 \tag{8}$$

Though the terms where $|T(i)| > \mathcal{T}$ may reduce the cumulative distance compared to a single-mode setting, an extremely large number of modes will result in excess modes only storing one/few tasks. These excess terms will increase, and the cumulative distance from $\theta^{\text{MTL}}$ will be greater in the multi-mode setting than the single-mode setting.

If $0 < N < \infty$ is optimized, then $\pi < 0$, given:

$$N \left[ \Sigma_t^{T(i)\big||T(i)|>\mathcal{T}} \frac{1}{|\theta|/N} \Sigma_{d=1}^{|\theta|/N} (\theta_{i,t,d} - \theta_d^{\text{MTL}})^2 + \Sigma_t^{T(i)\big||T(i)|\leq\mathcal{T}} \frac{1}{|\theta|/N} \Sigma_{d=1}^{|\theta|/N} (\theta_{i,t,d} - \theta_d^{\text{MTL}})^2 \right]$$

$$< \Sigma_t^{T(i)\big||T(i)|>\mathcal{T}} \frac{1}{|\theta|} \Sigma_{d=1}^{|\theta|} (\theta_{1,t,d} - \theta_d^{\text{MTL}})^2 + \Sigma_t^{T(i)\big||T(i)|\leq\mathcal{T}} \frac{1}{|\theta|} \Sigma_{d=1}^{|\theta|} (\theta_{1,t,d} - \theta_d^{\text{MTL}})^2 \tag{9}$$

For $|T(i)| \leq \mathcal{T}$, any sampled mode will be similarly distant from $\theta^{\text{MTL}}$, thus we can cancel this term on both sides.

$$\Sigma_t^{T(i)\big||T(i)|>\mathcal{T}} \frac{N^2}{|\theta|} \Sigma_{d=1}^{|\theta|/N} (\theta_{i,t,d} - \theta_d^{\text{MTL}})^2 < \Sigma_t^{T(i)\big||T(i)|>\mathcal{T}} \frac{1}{|\theta|} \Sigma_{d=1}^{|\theta|} (\theta_{1,t,d} - \theta_d^{\text{MTL}})^2 \tag{10}$$

This result shows that, compared to single-mode sequential learning, if we optimize the number of modes $N$, then we can minimize the cumulative distance with respect to $\theta^{\text{MTL}}$, and thus minimize the total task drift.

In other words, we conclude that optimizing the number of modes $N$ against capacity $|\theta|$ and tasks allocated per parameter $|T(i)|$ can outperform training on a single mode. If we increase $N$, then we can minimize the total task drift. If $N$ is too large, however, then the number of tasks allocated per parameter $|T(i)|$ decreases, and thus increases the number of redundant mode terms (and total task drift).

$$\pi = \Sigma_{i=1}^N \left[ \Sigma_t^{T(i)} \frac{1}{|\theta|/N} \Sigma_{d=1}^{|\theta|/N} (\theta_{i,t,d} - \theta_d^{\text{MTL}})^2 - \frac{1}{N} \Sigma_{t=2}^T \frac{1}{|\theta|} \Sigma_{d=1}^{|\theta|} (\theta_{1,t,d} - \theta_d^{\text{MTL}})^2 \right] < 0 \tag{11}$$

$$\Sigma_{i=1}^N \Sigma_t^{T(i)} \frac{1}{|\theta|/N} \Sigma_{d=1}^{|\theta|/N} (\theta_{i,t,d} - \theta_d^{\text{MTL}})^2 < \Sigma_{t=2}^T \frac{1}{|\theta|} \Sigma_{d=1}^{|\theta|} (\theta_{1,t,d} - \theta_d^{\text{MTL}})^2 \tag{12}$$

$\square$

