# OpenReview forum: "Multiple Modes for Continual Learning"
_ICLR.cc/2023/Conference — Submitted to ICLR 2023_

### Official Review · Reviewer_ydF1 · 2022-10-24

**Confidence:** 4
**Correctness:** 4
**Technical Novelty And Significance:** 2
**Empirical Novelty And Significance:** 2
**Recommendation:** 6

**Clarity, Quality, Novelty And Reproducibility:**



**Novelty**: As I explained before, the core ideas and main motivations of this work are already proposed. However, it may be the case that the authors were not simply aware of those works.


**Clarity & Quality**: Overall, the paper is clear. However, I believe in terms of organization and minor editorial matters, there is room for improvement.

**Strength And Weaknesses:**

### Strengths


- The paper is easy to follow, and the method is intuitive.


- The idea of continual learning with multiple modes and the trade-off analysis is interesting.



### Weakness:


- First, I should note that in terms of novelty and contributions, continual learning with multiple modes is not new. In fact, in [1] (see Alg. 4, 5), the authors build on the same idea of multiple modes motivated by neural network subspaces [2]. While in this work, there is an additional term to encourage diversity, the authors in [1] report the cosine regularization in [2] does not provide substantial additional benefits. Overall, while I suspect authors may not be aware of [1], in terms of novelty, I believe the core ideas of [1] are similar to this work.



- The proposed method, MOTA, has some limitations. First, it requires knowledge of task boundaries (to discourage drift). In addition, I believe the experiments do not cover the Class-IL setup. Also, I believe several other related baselines, such as [1] and [3], should have been included in benchmarks.



- [Minor; did not play a role in my decision] I believe the paper needs additional proofreading. More specifically, there are several typos (e.g., section 1, section 2), and on page 2, the paper refers to a figure on page 8.




**References**:


[1] Doan, Thang et al. “Efficient Continual Learning Ensembles in Neural Network Subspaces.” ArXiv abs/2202.09826 (2022).


[2] Wortsman, Mitchell et al. “Learning Neural Network Subspaces.” ICML 2021.


[3] Wen, Yeming et al. “BatchEnsemble: An Alternative Approach to Efficient Ensemble and Lifelong Learning.” ICLR 2020.





**Summary Of The Paper:**


The paper studies continual learning with multiple modes. The proposed method MOTA is motivated by mode connectivity and ensembles literature. MOTA aims to train $N$ diverse solutions on the first task, while for the next tasks, it uses the average/joint prediction to minimize the training loss and parameter drift. In addition, at the end of training each task, MOTA uses stored checkpoints to further optimize the drift. The authors show the effectiveness of MOTA on various tasks and instance incremental scenarios.




**Summary Of The Review:**

Overall, I find the idea of continual learning with multiple modes interesting, I believe the paper needs improvement regarding the organization/presentation. However, I think this is a good first step.




**Update (Post-Rebuttal)**
I would like to thank the authors for their response. Given that the authors have addressed most of my concerns and included new baselines, I would like to increase my score.

---

> ### Author Response · Authors · 2022-11-20
> **Specific Response**
>
> We have taken your suggestions to heart. We hope our response addresses your concerns, and that you will raise your score to acceptance.
>
> Firstly, we affirm that our method does not require knowledge of task boundaries (details [here](https://openreview.net/forum?id=7sWLxZBLPO5&noteId=BEHpixoAB2)). MOTA does not require task indices for switching between task-indexed parameters (like BatchEnsemble [a3]), nor does it require explicit task boundaries between batches of data. We also made proofreading corrections upon your suggestion.
>
> > “First, I should note that in terms of novelty and contributions, continual learning with multiple modes is not new. In fact, in [1] (see Alg. 4, 5), the authors build on the same idea of multiple modes motivated by neural network subspaces [2]. … I believe several other related baselines, such as [1] and [3], should have been included in benchmarks.”
>
> We extend on our discussion [here](https://openreview.net/forum?id=7sWLxZBLPO5&noteId=BEHpixoAB2) that we are the first to leverage the global geometry of a loss landscape changing with tasks without compromising the task-optimal properties of each subspace nor requiring any task-specific information.
> While BatchEnsemble [a3] and Doan et al. (2022) [a7] make use of multiple parameters, their limitations do not support both (I) retaining the properties of each subspace for different task settings, and (II) being able to access the properties (compatibly) of one subspace from another.
>
> BatchEnsemble [a3] requires the use of task-conditioned inputs to access task-specific parameters. Its strategy is to access parameters in task-indexed local subspaces and use one local subspace at a time for inference, but does not support the simultaneous access/use of all subspaces for the inference of a single test-time input. As all MOTA’s modes are trained using a joint loss (with respect to the joint probability distribution of all modes) per task, the modes are trained such that all modes can be used for inference, and thus access all the local subspaces without exclusion for any test-time point regardless of task. From your suggestion, we baseline against BatchEnsemble [a3] based on the author’s open-sourced implementation.
>
> Doan et al. (2022) [a7] trained N independent initialized parameters on all tasks across Algorithms 1-6. Though variations such as the use of replay and connectivity are presented, the basic setting of training N independently-initialized parameters on all tasks (Algorithm 3) resembles our ablation “Ensembles (independent seed)”, which MOTA outperforms. Moreover, MOTA does not assume uniform data exposure per parameter/mode. Each time a new task is added, the loss landscape changes. For each ensemble model in Doan et al. (2022) [a7], after learning a new task, the ensemble model would drift away from its previous position in the loss landscape - a comparison to this could be N x Single-Task Learners. By drifting away from this subspace, each ensemble model gradually loses the task-optimal properties of the subspace in which each ensemble model resided. MOTA’s optimization of task allocation per mode using a joint loss means each mode does not need to drift more than necessary per task (and drift is further hindered with a regularization term). As a result, MOTA is able to retain all the task-optimal subspaces, and does not overwrite them upon new tasks. We do not evaluate with Doan et al. (2022) [a7], in part as there is no available open source implementation to assist a quick reproduction during the rebuttal period, and in part because it does not appear to be peer-reviewed yet.
>
> Given our efforts in addressing your concern on novelty and implementing an additional baseline, we hope you will consider raising your score.
>
>
> [a3] Yeming Wen, Dustin Tran, and Jimmy Ba.  Batchensemble:  an alternative approach to efficient ensemble and lifelong learning. InInternational Conference on Learning Representations, 2020
>
> [a7] Doan, Thang et al. “Efficient Continual Learning Ensembles in Neural Network Subspaces.” ArXiv abs/2202.09826 (2022).

---

### Official Review · Reviewer_x9Gv · 2022-11-02

**Confidence:** 3
**Correctness:** 4
**Technical Novelty And Significance:** 3
**Empirical Novelty And Significance:** 2
**Recommendation:** 6

**Clarity, Quality, Novelty And Reproducibility:**

Clarity and quality are very nice.

Novelty seems mostly on the practical aspect.

**Strength And Weaknesses:**

Strength
1) Two regularization are proposed. One to max the difference of all models at 1st task. One to reduce param space drift between two tasks.
2) The MOTA work should be easy to enable and user friendly for many real world tasks.

Question
1) The two regularization both use the weighted factor when added to the loss. How robust would the model metric be when the weighted factor change?
2) As the results in Table 3 have base line approach from papers before or in 2018, are they the best base line model for now, or should the comparison add newer approach from papers after or in 2019?

**Summary Of The Paper:**

The authors consider the usage of more than one models for CL problem, and apply two regularization terms to maximize the difference of all models at 1st task and to minimize the parameter drift.

The paper claims better model quality when compared to other method.

**Summary Of The Review:**

Recommend accept, marginally over threshold. Major reasons for not higher score would include the extent for novelty and baseline model selection in the comparison, whether more advanced and newer methods are compared to MOTA.

---

> ### Author Response · Authors · 2022-11-20
> **Specific Response**
>
> We are grateful for your feedback in improving the paper! We reaffirm the novelty of our work as the first to leverage the global geometry of a loss landscape changing with tasks, with additions to the paper text (and further discussion [here](https://openreview.net/forum?id=7sWLxZBLPO5&noteId=BEHpixoAB2)). In continued support of our novelty, we added 4 post-2019 baselines, including two replay methods and two parameter isolation methods. To put into context, not only does MOTA continue to outperform all these baselines, but these methods have looser assumptions (e.g. making use of replay buffer data, access to task index) and are thus harder to beat. Based on your recommended changes, we hope you will raise your score to support acceptance.

---

### Official Review · Reviewer_EVp1 · 2022-11-04

**Confidence:** 2
**Correctness:** 3
**Technical Novelty And Significance:** 4
**Empirical Novelty And Significance:** 4
**Recommendation:** 8

**Clarity, Quality, Novelty And Reproducibility:**

The paper is, for the most part, very clear and easy to understand. It seems to be highly original, though "we are amongst the first to leverage the global geometry of the loss landscape for task adaptation," seems to suggest there may be other works with similar insights, and a more detailed comparison and claims of novelty would have been appreciated.

**Strength And Weaknesses:**

Strengths:
- Provides clear motivation to study multi-mode continual learning (Theorem 1).
- Shows creativity in the construction of its algorithm.
- Solid experimental result and its discussion.

Weaknesses:
- Theoretical justification for the specific steps of its algorithm (Section 3) lacking.
- In particular, motivations behind (1) the choice of parameter prior that maximizes the pairwise distance and (2) using random-weight-interpolated parameter in its loss function are unclear.
- The second paragraph of Section 3.2 (starting with "If a task has a high level of certainty, ...") is confusingly written and may benefit from an illustrative diagram or a figure.
- The paper claims that the algorithm does not depend on the knowledge of the task boundaries, and yet the specific implementation of the algorithm seems to depend on it (e.g. regularizing distance with parameter of previous task). Further clarification would be helpful.
- Conclusion is abrupt and does not provide any direction for future research.

**Summary Of The Paper:**

This paper introduces a novel method, Mode-Optimized Task Allocation (MOTA), for continual learning with superior backward and forward transfer performance as well as lower average task drift.

The paper first identifies the key weakness behind prior regularization-based continual learning methods: they anchor all future parameters to the first task observed, even when doing so degrades transfer performance. MOTA instead aims to leverage the global geometry of the parameter/loss space by instead optimizing $N$ different modes, each having a set of tasks allocated to it. It proves that when $N$ and the set of tasks per mode are optimized against parameter capacity $|\theta|$, the total task drift is lower than when $N=1$.

It describes the MOTA algorithm which first initializes the $N$ modes such that the pairwise distance among them are maximized, and updates the parameters for each task via checkpointed gradient updates such that the drift per mode is minimized.

The paper concludes after presenting empirical comparison of the transfer performance of MOTA with other baseline continual learning methods, which shows that its average accuracy is superior to all except multi-task learning.

**Summary Of The Review:**

Overall, the paper clearly identifies a limitation with prior regularization-based continual learning methods and suggests an alternative that is both theoretically and empirically well-supported and convincing. Although parts of its main algorithm lack clear motivation and its conclusion does not offer insights about its potential impact and directions for future research, it is an original and creative work that I strongly recommend.

---

> ### Author Response · Authors · 2022-11-20
> **Specific Response**
>
> Thank you for your support and appreciation of the insights in the paper! We have improved the paper further, affirming that we are the first to leverage the global geometry of a loss landscape changing with tasks, and even outperforming new baselines suggested by reviewers. We also confirm that we do not require any task-specific information at test-time (details [here](https://openreview.net/forum?id=7sWLxZBLPO5&noteId=BEHpixoAB2)). We also improved the text, e.g. extending the conclusion with future directions.

---

### Official Review · Reviewer_MaED · 2022-11-07

**Confidence:** 3
**Correctness:** 2
**Technical Novelty And Significance:** 2
**Empirical Novelty And Significance:** 2
**Recommendation:** 3

**Clarity, Quality, Novelty And Reproducibility:**

Clarity is very low - see above.

Quality seems low - only compares to one prior method (EWC), which is quite old and probably not SOTA.

Novelty seems small -there are several prior works on online learning with ensembles and sparse models.

Reproducibility seems reasonable.

**Strength And Weaknesses:**

The idea of using a mixture of sub-models ("modes"),, instead of a single large parameter model, seems intuitively reasonable, although is not very novel (there is a large prior literature on ensembles and mixture of experts, etc). Furthermore, the paper makes claims that I think are false, and is very hard to read.

One of the biggest problems is that the authors claim they do not need to know about task boundaries. Yet it is clear from algorothm 1 that the data is actually provided to the algorithm in chunks, with each chunk corresponding to a different task, and the algorithm is allowed to exploit this fact , by storing different parameter vectors for each mode i and task t, in the form theta(i,t). This seems contradictory. Furthermore, it seems the algorithm stores multiple copies of theta(i,t), one per epoch, to get theta(i,t,e). Thus the running time in step 12 is O(t E N) for each task t, resulting in a total complexity of O(T^2 E N). This seems expensive in time and space - more so than a simple replay buffer strategy (which they do not compare to).
Another problem is that their predictions are computed by using an unweighted average of the logits from each mode. What is the justification for this? If the modes are specialized on different tasks, why should they all get to vote equally?

The presentation is very unclear. For example, what does thm 1 even mean? theta_d^MTL is never defined. And why should we care about how much paramters drift or not? What does this have to do with predictive performance? Also, what assumptions are you making on the data stream (data distribution for each tasks)?

Many terms are either not explained (eg "EWC" is not defined), or very poorly explained. For example, the distinction between instance-IL, task-IL and class-IL is very unclear. It seems to have something to do with the label (output) space for each task, Y_t ( some partitioning of a fixed, larger ("fine grained") label set. ) . It seems the algorithm assumes ahead of time that every task will always be a C-way classification task, but the "meaning" of each label may change across tasks (eg for task 1, 0=dog, 1=cat, 2=horse; for task 2, 0=truck, 1=car, 2=airplane), which we can think of as a hierarchical two-part label, (task, class). So it seems once again that you do need to know the task boundaries.

The experiments are not very clear.
- In sec 4.0, What does it mean to say "We utilize resnets-{18,50,152} with {11, 181, 642 || 23, 528, 522 || 58, 164, 298} adjustable parametrs? What does the || notation mean? What are these adjustable parameters, and which ones are fixed?
- In sec 4.1, you compare to MTL as your "gold standard", but what exactly is this? (I assume you train a single big model with T output heads, each with C classes, in an offline way?)
- In sec 4.2, the meaning of fig 3 is unclear. It seems that the loss does not vary at all in the subspace of  mode 2, suggesting that a single mode may suffice. Also, for mode 1, the low loss region seems constant across tasks, but this might be an artefact of your regularizaton strategy. Once again, why should we care about how the parameter subspaces change? What does this have to do with predictive performance?
- In table 3, you only compare to somewhat older baseline methods (<=2018) -are these SOTA? And you only compare on 1 dataset (split cifar-100). Are the differenes between methods significant? (No error bars are reported!) Can the diferences be explained away by hyper-parameter tuning?
- tables 1-3 are all hard to read (bolding the best and second best might help, and/or focus on fewer metrics).





**Summary Of The Paper:**

The paper proposes a new heuristic approach to continual learning in which a set of models ("modes") are joinlty updated after each task, as in an ensemble method, but using a novel loss function, which encourages diversity. Some empirical gains (relative to some other heuristic methods) on some task incremental learning image classification benchmarks are shown.

**Summary Of The Review:**

The paper proposes some ad-hoc heuristics which are poorly explained and seem lacking in theoretical justification. The experimental results seem inconclusive.

---

> ### Author Response · Authors · 2022-11-20
> **Specific Response**
>
> Thank you very much for your feedback, we have taken them very seriously. We have addressed many of them [here](https://openreview.net/forum?id=7sWLxZBLPO5&noteId=BEHpixoAB2), improved the paper accordingly, and hope these addressed concerns will raise your score to acceptance.
>
> From our discussion [here](https://openreview.net/forum?id=7sWLxZBLPO5&noteId=BEHpixoAB2), we first address that we do not in fact need to assume task boundaries. MOTA does not require task indices for switching between task-indexed parameters, nor does it require explicit task boundaries between batches of data.
> In Algorithm 1, we subscript parameters with t, e to indicate the state of the parameters. However, we do not store a separate set of parameters per task per epoch for test-time. After each epoch, the N modes are overwritten with the optimal modes computed (Algorithm 1: line 13). We initialize N modes and end up with only N modes by test-time.
>
> Furthermore, the optimization of task allocation per mode, where we use a joint loss (based on the joint probability distribution of all modes per task) to update all modes, enables us to not only use an unweighted average, but it also enables us to leverage the global geometry of a loss landscape changing with tasks. Though prior work can make use of multiple ensemble models, they cannot (I) retain the properties of each subspace for different task settings, and (II) access the properties (compatibly) of one subspace from another.
>
> We have improved the presentation of our paper. We annotated Theorem 1 to show that the total task drift is lower in the multi-mode setting than single-mode setting. We also clarified the parameter count (e.g. 11,181,642 parameters for ResNet18). We do not freeze any additional parameters or make any changes to the parameters, we just directly load a pre-trained ResNet model from PyTorch. To not appear as though making biased conjectures for our setup, we also highlight Lange et al. (2021) [a5] for a description of instance/task-IL setups and multi-task learners (MTL), where classes are not hierarchical w.r.t. tasks. We also add 4 post-2019 baselines, which we also continue to outperform.
>
>
> [a5] M. De Lange et al., "A Continual Learning Survey: Defying Forgetting in Classification Tasks," in IEEE Transactions on Pattern Analysis and Machine Intelligence, vol. 44, no. 7, pp. 3366-3385, 1 July 2022

---

### Author Response · Authors · 2022-11-20
**General Response**

We thank the reviewers for their feedback and suggestions. We highlight overarching concerns listed across all reviewers here, and address unique concerns individually.

**A. Reviewing the assumptions in continual learning**

We reaffirm that we do not make use of task-specific information. This includes switching between task-indexed parameters, or requiring batches of data to have explicit task boundaries. To not appear as though making biased conjectures about the continual learning setting to our advantage, we also defer to the peer-reviewed survey from Lange et al. (2021) [a5] for details on the instance-IL [a6: Sec1] and task-IL setups [a5: Sec2], single-task and multi-task learner baselines [a5: Sec6.1], and other assumptions [a5: Sec7].

**What constitutes a task?** We refer to a task as a new batch of data pertaining to a new sub-population/domain (but same label set and output space), new label set (but same output space), or different output space. For example, the Instance-IL tasks in Table 1a and the DomainNet tasks in Table 2b use the same labels between classes. When evaluating on a different sub-population but sharing the same coarse labels (Table 1a), or evaluating on a different domain but sharing the same labels (Table 2b), we do not have a separate label set (nor do we assign any task index or store any replay buffer) for each new domain/sub-population.

**Task boundaries.** We have now amended to refer to this as “task-specific information” and state in the paper that MOTA does not use task-specific information to switch between task-indexed parameters, nor does it need batches to possess task boundaries. In-line with regularization-based methods such as EWC, we do not require any task-specific information to condition the parameters of MOTA. We receive a new batch of data to update the N modes, but we do not store N*T modes (i.e. the N modes from the previous task are over-written). We do not select a set of modes based on task index at test-time. When the modes have not been initialized yet, on the first batch of data (task1), we perform Algorithm 2 (now renamed to initialize_parameters). Subscripts t, e in Algorithms 1-3 indicate the latest state of the same over-written mode. Given parameters are not conditioned on task indices or buffers, we are able to perform online continual learning with batches that do not possess task indices, such as new sub-populations or domains of the same label set.

---

> ### Author Response · Authors · 2022-11-20
> **General Response (cont'd)**
>
> **B. Affirming contribution:  the first to leverage the global geometry of a loss landscape changing with tasks without compromising the task-optimal properties of each subspace nor requiring any task-specific information**
>
> We welcome the clarification questions raised on global geometry, we will now have an extended discussion on the insight that went into MOTA as well as solidify our claims on novelty.
>
> Firstly, it should be stated that just because multiple parameters are used, does not mean that a method leverages the global geometry of the parameter space. Ensemble methods (e.g. [a7]) indeed have multiple parameters, and train the N models on all tasks and update all parameters. However, the loss landscape changes after each task. Unlike conventional loss landscapes, in task adaptive settings such as continual learning, the loss dimension of the loss landscape changes as we switch evaluation from one task to another task. As a result, when these ensemble methods update their parameters with respect to a new task, they overwrite the properties of the previous task parameter’s local subspace, and the ensemble model drifts to a new subspace. They lose the properties of the local subspace that enabled it to perform well on previous tasks. Having lost the properties of the subspace for specific tasks, it fails to leverage the global properties that different subspaces have for different tasks. On the other hand, ensembles (e.g. BatchEnsemble [a3]) can be constructed such that N models are trained on N respective tasks (1 task per model). Other than the fact that this requires task indexing to select the known model for a task at test time, this also does not leverage the global geometry of the loss landscape. Each model and its subspace is independent from each other (the model in one subspace does not affect the model in another subspace). While the previous type (e.g. [a7]) lost properties of local subspaces, this type (e.g. [a3]) is fully reliant on only utilizing local subspaces. To summarize, leveraging the global geometry is not just a matter of multiple parameters. It is also about (I) retaining the properties of each subspace for different task settings, and (II) being able to access the properties (compatibly) of one subspace from another.
>
> To summarize, prior work either anchors to the local subspace of the first task, anchors each task to its specific local subspace, or anchors the entire parameter space to the last seen task. As such, we are the first to leverage the global geometry of a loss landscape changing with tasks without compromising the task-optimal properties of each subspace nor requiring any task-specific information.

---

> > ### Author Response · Authors · 2022-11-20
> > **General Response (cont'd)**
> >
> > **C. Added comparison to other baselines**
> >
> > We used EWC as the primary baseline throughout the paper as it is considered a state-of-the-art regularization-based method. Many other state-of-the-art methods (e.g. replay or parameter isolation) may change assumptions in their problem setting (e.g. assuming capacity can expand, require clear task boundaries or task indices). Continual learning papers continue to use EWC as a baseline. It is a good baseline in any future work in continual learning, as it does not require capacity expansion or separate buffers or architectural changes or task indexing.
> >
> > Based on reviewer feedback, we add two post-2019 replay baselines (La-MAML [a1], GPM [a2]), and two post-2019 parameter isolation baselines (BatchEnsemble [a3], WSN [a4]), in-line with the open-source implementations provided by the respective authors. Replay and parameter isolation baselines have relaxed assumptions, as they can make use of task indices, buffers, etc. The inclusion of more non-regularization methods dilutes the sharp comparison we wish to make, as the assumptions/tradeoffs that each method make do not come across in the table, so it is still best to compare our work to regularization-based methods. In addition to the two ensemble ablations, BatchEnsemble is also a baseline for the use of ensembling in continual learning. MOTA continues to outperform all baselines.
> >
> > **D. Presentation improvements**
> >
> > We accepted various suggestions made by reviewers on improving clarity of the text and uploaded a revised manuscript (with key changes highlighted).
> >
> > **References**
> >
> > [a1] Gunshi Gupta, Karmesh Yadav, and Liam Paull. Look-ahead meta learning for continual learning.Advances in Neural Information Processing Systems, 33:11588–11598, 2020.
> >
> > [a2] Gobinda Saha, Isha Garg, and Kaushik Roy. Gradient projection memory for continual learning. In International Conference on Learning Representations, 2021.
> >
> > [a3] Yeming Wen, Dustin Tran, and Jimmy Ba.  BatchEnsemble:  an alternative approach to efficient ensemble and lifelong learning. InInternational Conference on Learning Representations, 2020
> >
> > [a4] Haeyong Kang, Rusty John Lloyd Mina, Sultan Rizky Hikmawan Madjid, Jaehong Yoon, Mark Hasegawa-Johnson, Sung Ju Hwang, and Chang D Yoo.   Forget-free continual learning with winning subnetworks.   InInternational Conference on Machine Learning,  pp. 10734–10750.PMLR, 2022.
> >
> > [a5] M. De Lange et al., "A Continual Learning Survey: Defying Forgetting in Classification Tasks," in IEEE Transactions on Pattern Analysis and Machine Intelligence, vol. 44, no. 7, pp. 3366-3385, 1 July 2022
> >
> > [a6] Lomonaco, V. & Maltoni, D.. (2017). CORe50: a New Dataset and Benchmark for Continuous Object Recognition. Proceedings of the 1st Annual Conference on Robot Learning, in Proceedings of Machine Learning Research 78:17-26
> >
> > [a7] Doan, Thang et al. “Efficient Continual Learning Ensembles in Neural Network Subspaces.” ArXiv abs/2202.09826 (2022).

---

### Decision · Program_Chairs · 2023-01-20

**Decision:**

Reject

**Justification For Why Not Higher Score:**

The major concern about this paper is on comparison with the original EWC mainly, which is clearly too old as a baseline for regularisation based continual learning. Therefore it is difficult to understand how significant the gain of MOTA is.

**Justification For Why Not Lower Score:**

N/A

**Metareview: Summary, Strengths And Weaknesses:**

The paper proposes the MOTA for continual learning. The idea is to train jointly a set of models for each task (like an ensemble) with an objective that encourages diversity, and then continual learning is done by coordinating the updates of the ensemble to achieve minimum model drift. Empirical analysis on task incremental learning image classification benchmarks shows MOTA achieves improvements over baselines.

Initial reviews had mixed opinions, overall they felt the perspective of balancing and coordinating networks in an ensemble for continual learning is very interesting, with one reviewer very supportive for acceptance. But others had doubts on novelty of the approach and the selection of baselines (EWC as a method published 5 years ago). The authors updated the manuscript with more baselines in one of their experiments (table 3). However, other results still use EWC as the only baseline. Furthermore, comparisons/visualisations of other ensemble-based continual learning method is missing.

For future submissions, I would suggest the authors to include more results relevant to ensemble-based continual learning in their analysis.

PS: purely looking at the score, this paper might seem to be borderline. In the end I conclude it is not borderline. See below for explanations.

**Summary Of Ac-Reviewer Meeting:**

This paper might be viewed as a borderline paper mainly because of one 8/10 (originally 10/10) review (EVp1). Other reviewers are not particularly excited about this paper.

After communications with reviewers including reviewer EVp1, this reviewer admitted that the original score they gave was not well supported. I personally believe the current 8/10 score is not well supported by the actual review text either (e.g., the review mentioned issues of this paper and those issues are not addressed in the current revision). Considering the actual review text, I see the review by EVp1 as with "good will of support but not grounded well by evidence". This reviewer didn't provide enthusiastic support for acceptance after we had this communication exchange.

In summary, I consider EVp1 review score to be less well-calibrated. Combined with other reviews, I conclude that this paper is not borderline.